# Evaluation of Biological Activity of Natural Compounds: Current Trends and Methods

**DOI:** 10.3390/molecules27144490

**Published:** 2022-07-13

**Authors:** Carlos Barba-Ostria, Saskya E. Carrera-Pacheco, Rebeca Gonzalez-Pastor, Jorge Heredia-Moya, Arianna Mayorga-Ramos, Cristina Rodríguez-Pólit, Johana Zúñiga-Miranda, Benjamin Arias-Almeida, Linda P. Guamán

**Affiliations:** 1Escuela de Medicina, Colegio de Ciencias de la Salud Quito, Universidad San Francisco de Quito USFQ, Quito 170901, Ecuador; cbarbao@usfq.edu.ec; 2Centro de Investigación Biomédica (CENBIO), Facultad de Ciencias de la Salud Eugenio Espejo, Universidad UTE, Quito 170527, Ecuador; saskya.carrera@ute.edu.ec (S.E.C.-P.); rebeca.gonzalez@ute.edu.ec (R.G.-P.); jorgeh.heredia@ute.edu.ec (J.H.-M.); arianna.mayorga@ute.edu.ec (A.M.-R.); cristina.rodriguez@ute.edu.ec (C.R.-P.); johana.zuniga@ute.edu.ec (J.Z.-M.); benjaarias94@gmail.com (B.A.-A.)

**Keywords:** natural product, bioactive compounds, antimicrobial, antioxidant

## Abstract

Natural compounds have diverse structures and are present in different forms of life. Metabolites such as tannins, anthocyanins, and alkaloids, among others, serve as a defense mechanism in live organisms and are undoubtedly compounds of interest for the food, cosmetic, and pharmaceutical industries. Plants, bacteria, and insects represent sources of biomolecules with diverse activities, which are in many cases poorly studied. To use these molecules for different applications, it is essential to know their structure, concentrations, and biological activity potential. In vitro techniques that evaluate the biological activity of the molecules of interest have been developed since the 1950s. Currently, different methodologies have emerged to overcome some of the limitations of these traditional techniques, mainly via reductions in time and costs. These emerging technologies continue to appear due to the urgent need to expand the analysis capacity of a growing number of reported biomolecules. This review presents an updated summary of the conventional and relevant methods to evaluate the natural compounds’ biological activity in vitro.

## 1. Introduction

According to the World Health Organization (WHO), 80% of the global population uses medicinal-plant-based medicine to alleviate or cure diseases [1]. In addition, although estimates vary depending on what is considered a natural-product-derived drug, it is safe to say that up to 50% of currently marketed drugs owe their origins to natural products [2]. New molecules from natural resources with potential bioactivity are reported every day; however, only a few of these molecules are evaluated for their suitability for use as drugs [3]. Identifying bioactive compounds (hits or leads) is the initial step for drug discovery. Therefore, it is necessary to select suitable bioassays to evaluate both the activity against the disease and the potency. For this purpose, target-based screening is mainly used to identify compounds that modulate the activity of a target involved in a disease. This screening involves different in vitro biological assays designed to measure primary activities, selectivity, cellular toxicity, and physiologically relevant activity. The initial phases of a target-based screening cascade typically employ a range of in vitro assays, especially high-throughput screening (HTS); however, the study is more expensive and time-consuming [4,5]. In the first instance, the assays can be selected considering that structurally similar compounds have similar biological activity; however, this cannot always be carried out, especially when working with natural compounds or natural extracts. 

Extracts from various plants are commonly used in traditional medicine, either alone or in combination, but in many cases only a few of them are evaluated for their biological activity. However, due to inadequate fractionation processes or the degradation of active compounds during separation, it is not always easy to identify the molecules responsible for the activity of these extracts. Since obtaining an isolated compound very often requires infrastructure and specialized personnel and is expensive, it is challenging to offer pharmacological alternatives of this nature to people with low incomes. In addition, since medicinal herbs can be grown locally at a reasonable cost, natural extracts remain an option for treating some diseases in populations with limited resources or living in remote areas [6].

An important advantage of using crude extracts vs. isolated molecules is the presence of molecules in the extract that can interact synergistically with the bioactive compound, potentiating its beneficial effect [7]. However, despite their potential as accessible treatment options and as sources of bioactive molecules for drug discovery, it must be highlighted that similar to other pharmacological alternatives, natural extracts can also present adverse effects that should be considered and evaluated. Having many assays on hand to discard or confirm activities is critical when looking for bioactive compounds because the aim while looking for therapeutic agents is to identify an acceptable technique that can screen the source material for bioactivity. Within this context, the current review presents some of the most common in vitro assays currently used for the identification of the pharmacological activity of bioactive compounds or natural extracts and that allow the preliminary identification of their biological potential and possible targets. Additionally, each section highlights the advantages and disadvantages of these methods, guiding the reader to the selection of the best assays based on the type of sample and available resources. Finally, alternatives or future trends to these approaches are also included in each section. 

## 2. Cytotoxicity Activity in Cultured Mammalian Cells

In the early stages of drug development, extensive toxicity screening is essential [8,9]. Animal studies involve high costs and are often restricted by differential responses due to the physiological differences between species and limitations in test feasibility. Alternatively, in vitro cytotoxicity assays are advantageous in preclinical studies due to their eligibility, cost-effectiveness, and reproducibility. Natural compounds have become particularly relevant in identifying safer and more effective treatments [10].

To evaluate the cytotoxicity in mammalian cells, it is critical to select the proper cell line for each particular experiment, considering the relevant species, specific organs, and chosen route of administration. Multiple cell lines are available for in vitro testing, including immortalized cell lines, primary cultures, and stem cells. Each cell line has requirements that need to be defined before the experiment to ensure a proper understanding of the potential mechanisms of toxicity [11,12]. 

A wide range of in vitro assays are currently available for cytotoxicity testing [13,14]. While direct cytotoxicity assays focus on detecting a loss of membrane integrity associated with cell death, cell viability assays are developed to measure the activity related to cellular maintenance and survival. Additionally, other assays allow the direct and indirect quantification of changes in the population at specific phases of the cell cycle and provide information on the mechanism of cell death [15]. Different approaches are employed in parallel to better understand the cytotoxicity mechanisms, such as the terminal deoxynucleotidyl transferase dUTP nick end labeling (TUNEL) assay and the comet assay for the analysis of DNA fragmentation [16]; the determination of the activation of apoptosis-related caspases [17,18]; or the detection of the relative level of telomerase activity (TRAP, telomerase repeat amplification protocol) [19,20]. Determining the cytotoxicity against tumor cells; detecting the rate and regulation of cell migration; and analyzing anchorage-independent proliferation, chemotaxis, and invasion are essential factors usually evaluated using colony-forming assays in soft agar and scratch assays using dual-chamber systems [21,22,23]. 

In short, each assay has its limitations. Although the 3-[4,5-dimethylthiazol-2-yl]-2,5 diphenyl tetrazolium bromide (MTT) assay has been generally accepted as the gold standard in cytotoxicity testing, this method is not always the most pertinent due to interference with the test compounds, particularly with natural extracts [24,25]. Therefore, before selecting an appropriate methodology, different assays should be compared and more than one should be used when possible. The cost, reliability, timing, user-friendliness, and equipment requirements should also be considered [26]. Table 1 presents the advantages and limitations of these assays. 

Although two-dimensional mammalian monocultures stemming from specific cell types are widely used based on their reproducible and rapid growth, high productivity levels, ease of data interpretation, and value, the artificial nature of the culture environment presents limitations in drug safety and efficacy evaluation [47,48]. In this sense, even though improved versions of the classical reagents are being developed, the field’s current focus is shifting towards co-culture systems, human organoids, and other sophisticated three-dimensional culture models that collect more physiologically relevant data and represent methods that could better connect traditional cell culture and in vivo models [49,50,51] (Figure 1).

From the array of cell cultures available for in vitro testing that offer diverse degrees of intricacy and similarity to the in vivo setting, organotypic cultures are tissue slices that maintain cell interactions and the extracellular matrix composition of the original tissue and tissue function [52,53,54,55]. However, this system lacks intercommunication with the circulatory and immune systems and is inadequate for medium- to high-throughput analysis [11]. Three-dimensional spheroids and organoids self-organize into organ-specific structures that accurately replicate paracrine and direct intercellular interactions [56,57,58]. While spheroids are usually made from cell lines and offer lower complexity, organoids are derived from the stem cells of different origins and resemble the original tissue in the structure, histologically and genetically [59,60]. Tumor organoids are particularly relevant, since these systems provide suitable platforms to recapitulate the complex tumor microenvironment and its heterogeneity, allowing the study of chemical and metabolic gradients and mechanisms of resistance [49,61].

The evidence indicates that microfluidic devices are gaining traction in the area of cytotoxicity assays [62,63]. Compared to static conditions, microfluidic systems can reproduce the specific flow, temperature, pressure, and chemical gradients of the in vivo systems [64,65,66]. Thus, they can reconstruct the continuous renewal of nutrients, gasses and toxic wastes, migration, and microcirculation. These systems also support longer culture times and drug treatments that are more pharmacologically significant [67].

## 3. Antihyperglycemic Activity

Diabetes is a global health disease affecting 422 million people worldwide [68]. This disease is characterized by elevated blood glucose levels, which if untreated leads to severe multi-organ failure and 1.5 million deaths each year. Antihyperglycemic agents are a heterogeneous group of molecules obtained via chemical synthesis or isolation from natural sources that lower the glucose concentration in the blood or prevent its increase [69].

The methods used to evaluate the in vitro antidiabetic properties of natural compounds are summarized in Table 2 and can be classified into two major groups: (i) assays based on the inhibition of isolated enzymes involved in the regulation of blood glucose levels and (ii) assays used to measure major cellular processes that directly alter glucose levels, mainly glucose uptake and insulin secretion [70]. The first group includes enzymes that catalyze the breakdown of poly- and oligosaccharides such as α-amylase and α-glucosidase, respectively (Table 2). The inhibition of the mentioned enzymes and others with a similar role in carbohydrate digestion is considered antidiabetic. The reduction in the glucose concentration available to absorb in the intestine prevents a further increase in blood glucose [71]. The α-amylase and α-glucosidase inhibition assays are reactions of commercially available enzymes and substrates in the optimal conditions (buffer, pH, cofactors), allowing the detection of the reaction product(s) [72].

The most common substrate used to measure α-amylase is starch. The method is based on the reaction of starch with dinitrosalicylic acid (DNS), which reacts with reducing sugars, producing 3-amino-5-nitrosalicylic acid, which is measured spectrophotometrically at 540 nm. Most α-glucosidase assays rely on the spectrophotometric detection of p-nitrophenol liberated after the hydrolysis of *p*-nitrophenyl-α-d-glucopyranoside (pNPG), which can be measured at 400 nm. Dipeptidyl peptidase IV (DPP4) and tyrosine phosphatase 1B (TP1B) are involved in the indirect regulation of glucose levels by modulating insulin secretion (DPP4) [73] and signaling (TP1B) [74], respectively (Table 2). DPP4 is a serine exopeptidase that cleaves different peptides, including GLP-1, a major regulator of insulin secretion in response to glucose [75]. Thus, inhibiting DPP-4 in vivo increases the availability of GLP-1 and insulin secretion, reducing blood glucose [76]. TP1B negatively regulates insulin and leptin signaling by dephosphorylating the insulin receptor (IR) and its downstream signaling components [77,78]. The inhibition of TP1B releases insulin signaling from TP1B-mediated dephosphorylation and allows insulin downstream signaling [72]. The second group includes assays designed to measure the potential inhibitory effects of different molecules on relevant cellular processes controlling blood glucose levels, such as glucose uptake and insulin secretion (Table 2). The glucose uptake assay is based on the internalization of a labeled glucose analog that cannot be fully utilized because of its modification. It accumulates inside the cells, facilitating its detection. The output generated by the accumulation of labeled analogs is proportional to the glucose uptake and can be detected and quantified using standard equipment such as fluorescence or bioluminescence readers [79,80] or fluorescence-activated cell sorting (FACS) [81]. These assays can be performed in mammalian cell lines, given the importance of measuring the physiologically relevant effects. Some studies report using yeast cells as an alternative to mammalian cell lines [82]. For example, a recent study described a label-free method to measure glucose uptake in yeast cells using pHluorin, a genetically encoded pH-sensitive green fluorescent protein [83]. In general, insulin secretion assays are performed using ß-cells isolated from pancreatic or islet cell cultures. The cells are stimulated by glucose and incubated with the compound or plant extract to measure the insulin secretion modulation effect [84]. After being released from cells, insulin can be measured via radioimmunoassay [85,86] or ELISA. Recently, a luminescent alternative to detect insulin was described by Hager et al. and Kalwat et al. [87,88].

**Table 2 molecules-27-04490-t002:** Antihyperglycemic activity.

Method Name	Assay Type	Description	Detection (Output)	Advantages	Disadvantages	Ref.
α-amylase inhibition	Isolated enzymes	Measurement of the ability of novel molecules to inhibit the activity of α-amylase	Colorimetric and fluorometric assays	- Rapid - Simple - Cheap - Easy to escalate/automate	- Specialized lab equipment required - Isolated enzymes required	[89]
α-glucosidase inhibition	Isolated enzymes	Measurement of the ability of novel molecules to inhibit the activity of α-glucosidase	Colorimetric, fluorometric and bioluminescent assays	- Rapid - Simple - Easy to escalate/automate	- Specialized lab equipment required - Isolated enzymes required	[89,90]
Dipeptidyl peptidase IV inhibition	Isolated enzymes	Measurement of the ability of molecules to inhibit the activity of dipeptidyl peptidase IV	Colorimetric, fluorometric, immunoassay	- Rapid - Simple - Easy to escalate/automate	- Specialized lab equipment required - Isolated enzymes required	[89,90]
Tyrosine phosphatase 1B inhibition	Isolated enzymes	Measurement of the ability of molecules to inhibit the activity of tyrosine phosphatase 1B	Colorimetric, fluorometric	- Rapid - Simple - Easy to escalate/automate	- Specialized lab equipment required - Isolated enzymes required	[74,77]
Glucose uptake	Cell-based	Measurement of the ability of molecules to modify glucose uptake into cells	Colorimetric, fluorometric, bioluminescent assays	- Physiologically meaningful - Various cell models - Easy to escalate/automate	- Specialized lab equipment required - Highly trained personnel required	[91,92]
Insulin secretion	Cell-based	Measurement of the ability of molecules to modulate insulin secretion	Bioluminescent assay, immuno-/radioimmunoassay	- Physiologically meaningful - Easy to escalate/automate	- Highly trained personnel required - Specialized lab equipment required	[87,88]

Despite continuous improvements in measuring glucose and glucose-associated processes, a relevant challenge is to fully understand the physiological and pathological roles of blood glucose levels and their impact on health conditions and disease. In vitro methods are critical for discovering natural compounds with antidiabetic activity. Although diabetes and other health issues associated with high blood glucose levels can be treated using available antidiabetics, given the vast chemical diversity of natural products with unknown but potentially beneficial effects, the evaluation of the antidiabetic activity of molecules obtained from natural sources is a very relevant research topic. Some parameters that are actively being improved to allow the efficient prospection of potential candidates for developing novel antidiabetic drugs from natural sources are: (i) increasing the sensitivity and robustness of the assays; (ii) reducing the laborious steps needed for the preparation of samples and cell extracts; and (iii) increasing the throughput of the assays.

## 4. Anti-Inflammatory Activity

Inflammation is a protective response of a given organism’s immune system against harmful external agents, such as pathogens, toxins, or irritants [93]. This mechanism aids in the recovery from infections, disease, and tissue damage, favoring the healing process [94]. Inflammatory responses include the activation of macrophages by pro-inflammatory mediators such as lipopolysaccharide (LPS), interleukin-1β (IL-1β), interferon-γ (IFN-γ), and the nuclear factor kappa B (NF-κB), which trigger the cyclooxygenase (COX) and lipoxygenase (LOX) pathways and the production of nitric oxide (NO), tumor necrosis factor-α (TNF-α), and interleukin-6 (IL-6) primarily [95,96,97]. All of these pro-inflammatory mediators are studied to identify the anti-inflammatory properties of natural molecules in vitro. In terms of the anti-inflammatory treatments, non-steroidal drugs have been shown to inhibit the production of arachidonic acid and to lower prostaglandin levels, contributing to reducing pain and inflammation [98]. However, their use is associated with multiple side effects, including stomach ulcers, indigestion, headaches, allergic reactions, and increased cardiovascular conditions [98]. Fortunately, the incredible variety of phytoconstituents present in plants such as flavonoids, alkaloids, saponins, coumarins, anthraquinones, saccharides, glucosinolates, tannins, phenolic acids, and nitrile glycosides are excellent sources for drug development due to their vast biological activities [99]. Specifically, flavonoids [100,101], anthocyanins [102], and some polyphenols [103] have shown anti-inflammatory properties. Similarly, secondary metabolites, such as atranorin from lichens [104] and sulfated polysaccharides from the brown alga *Sargassum cristaefolium* [105], inhibit the inflammatory process [104]. These few examples highlight the incredible potential of natural extracts for developing anti-inflammatory pharmaceuticals. For a complete review of natural active molecules with anti-inflammatory potential, refer to Wang and Zeng [106]. A simple way of measuring inflammation is through the overproduction of nitric oxide (NO), which is associated with tissue toxicity, several inflammatory conditions, and carcinomas [58]. Easy, cheap, and rapid quantification of nitric oxide levels directly from a given compound or from a cell culture subjected to an inflammatory stimulus is possible through the Griess assay (Figure 2) [95,107,108,109]. Thus, a compound inducing low NO levels has anti-inflammatory potential. However, this method has drawbacks, including its variable sensitivity, low detection levels, and the interference of some compounds with the Griess reaction assay [110,111]. Alternatives to the Griess assay for NO detection are covered by Bryan and Grisham [112].

Other approaches, such as the enzyme-linked immunosorbent assay (ELISA) and qRT-PCR, have been extensively employed to determine the decreases in pro-inflammatory enzyme levels and their changes in expression, respectively (Figure 2) [94]. The advantages of ELISA are its simplicity, high specificity, and sensitivity. However, on the other hand, it is a time-consuming process and involves high costs associated with the antibodies [113]. In the case of qRT-PCR, the advantages include its high sensitivity and relatively high throughput. In contrast, its disadvantages are associated with its increased complexity, variable reproducibility, and the high cost of the equipment and reagents [114]. Additionally, qRT-PCR measures the mRNA levels but does not provide information about the overall enzyme levels. Alternative methods to measure inflammation include the nuclear factor kappa B (NF-κB) luciferase assay [108]; Western blotting assay on COX-2 and inducible nitric oxide synthase (iNOS) expression [115]; immunofluorescence staining of NF-κB p65 and iNOS [96]; ferrous oxidation−xylenol orange (FOX) assay for the determination of lipid hydroperoxides [116]; and protein denaturation inhibitory activity, heat-induced hemolysis, and lipoxygenase inhibitory activity assays [117]. However, these are less common during the early stages of anti-inflammatory activity identification.

It is essential to identify the best method for an investigation based on its advantages and limitations. Currently, researchers employ a combination of methods (e.g., NO inhibition assay and pro-inflammatory enzyme quantification) to obtain more information about the potential anti-inflammatory properties of different compounds. Figure 2 details the most common strategies used to establish the anti-inflammatory potential of natural molecules in vitro.

## 5. Analgesic Activity

Pain is a symptom of many diseases that require analgesic treatment [118]. An analgesic is a drug used to relieve pain without loss of consciousness [119]. Analgesics can act on the central or peripheral nervous system [120]. There are two main groups of drugs used to reduce pain: (i) non-steroidal anti-inflammatory analgesics, which alleviate pain by reducing local inflammatory responses; (ii) opiate analgesics, which are highly effective because they reduce but do not block the perception of the central nervous system and even induce sedation, transforming pain into a non-bothersome sensation [121,122]. There are several types of opioid receptors: µ-receptors, which when stimulated, the typical effects of narcotic analgesics are produced; κ-receptors, which are responsible for spinal analgesia [123], and when the drug acts on these receptors, it does not have addictive effects but causes unpleasant hallucinations; δ-receptors, which are responsible for cardiovascular manifestations [118,124].

Radioligand binding assays are used to assess the affinity and selectivity of the new molecules for specific receptors to evaluate the potential analgesic activity of new compounds. The three experimental types of radioligand binding assays are saturation, competitive, and kinetic assays [125]. The competitive assay is the most widely used, which studies equilibrium binding at a fixed radioligand concentration and with different concentrations of an unlabeled competitor [126]. In Table 3, the in vitro methods for analgesic activity are described.

There is a good correlation between the in vivo pharmacological potency of opiate agonists and antagonists and their ability to displace radiolabeled compounds such as naloxone, a substance that binds to opiate receptors [127]. The correlation mentioned above can be exploited to evaluate new compounds. As shown in Figure 3, these experiments are performed on animal brain membranes, which have a high density of the receptors of interest; in the assay, different concentrations of the test compound are evaluated against a fixed concentration of specific radioligands [128,129,130]. These assays help identify compounds that recognize the same binding site as known radiolabeled ligands. The results can be quantified and expressed as a percentage displacement of the radioactive compound and as IC_50_ value [119].

Enzyme assays are also used to determine analgesic activity (Table 4). The inhibition of enkephalinases has shown antinociceptive properties [126]. The determination of the inhibition is carried out via fluorescence, for which the dansyl-d-Ala-Gly-Phe(*p*NO_2_)-Gly (DAGNPG) molecule is used as a selective enzymatic substrate and the enkephalinase breaks the Gly-Phe(*p*NO_2_) DAGNPG peptide bond, which causes an increase in fluorescence [121].

**Table 3 molecules-27-04490-t003:** Methods used for the determination of the analgesic activity of molecules based on the displacement of radioligands.

Method Name	Assay Type	Description	Advantages	Disadvantages	Ref.
Displacement of radioligands at receptors	^3^H-N binding assay	Radiolabeled ligand: ^3^H-N Type of receptor: opiate Tissue culture: rat brain	- Sensitive method - Good robustness - Precise determination of ligand binding sites and affinity	- Other multiple opioid receptors are not considered - High cost - Hazards of handling high levels of radioactivity - Requires a certain level of expertise	[118,119]
	^3^H-DHM binding assay	Radiolabeled ligand: ^3^H-DHM Type of receptor: µ-opiate Tissue culture: rat brain			[124]
	^3^H-B binding assay	Radiolabeled ligand: ^3^H-B Type of receptor: κ-opiate Tissue culture: guinea pig cerebellum			[119,131]
	^3^H-NCNH_2_ binding assay	Radiolabeled ligand: ^3^H-NCNH_2_ assay Type of receptor: nociceptin Tissue culture: rat brain			[131]
	^35^S-GT binding assay	Radiolabeled ligand: ^35^S-GT Type of receptor: cannabinoid Tissue culture: human brain			[131,132]
	^3^H-R binding assay	Radiolabeled ligand: ^3^H-R Type of receptor: vanilloid Tissue culture: rat brain			[133,134]

^3^H-N: ^3^H-naloxone; ^3^H-DHM: ^3^H-dihydromorphine; ^3^H-B: ^3^H-bremazocine; 3H-NCNH_2_: ^3^H-nociceptin amide; _3_H-R: ^3^H-resiniferatoxin; ^35^S-GT: ^35^S-guanosine-5′-*O*-(3-thio) triphosphate.

Some of the current widely used analgesic treatments such as the non-steroidal anti-inflammatory drugs (NSAID) (e.g., acetylsalicylic acid) are associated with certain adverse effects, including gastrointestinal irritation and bleeding, fluid retention, and prolonged bleeding times [125]. Similarly, opioids, the current “gold standard” medication for nociceptive pain, also cause dose-limiting side effects including constipation, potentially lethal respiratory arrest, and drowsiness and are associated with the development of tolerance, dependence, and addiction [137]. Medicinal plant extracts offer a diverse source of new molecules with analgesic potential; however, it is essential to evaluate their adverse effects. Ideally, these natural compounds must be highly selective for a target that is specifically expressed in the nociceptive system, minimizing the risks of side effects [128]. Morphine, steroids, salicin, nicotine, artemisinin, atropine, pilocarpine, capsaicin, quinine, scopolamine, and captopril are a few examples of naturally derived molecules with analgesic properties [135]. Moreover, other plant-derived molecules such as monoterpenes, sesquiterpenes, and phenylpropanoids [136] and some venom peptides from animals such as spiders [138,139] have shown promising analgesic potential as well.

Overall, the assays presented above allow the screening of the analgesic activity of compounds in new extracts or natural products and encourage the isolation of the main active molecules present to relate them to their biological responses. Furthermore, knowing the chemical structure of the isolated molecules would allow an understanding of the possible biomolecular targets and their mechanisms of action to mitigate pain. Future research will need to focus on developing and improving multiple radio- and fluorescent ligands for studies with other types of specific binding, such as Na_V_ channel isoforms, the σ-opioid receptor, the ε-opioid receptor, and the bradykinin B1 receptor, among others.

## 6. Anticoagulant Activity

The coagulation process is usually described as a “waterfall” or “cascade”, referring to the steps involved in the development of a blood clot following injury by activating a cascade of proteins known as clotting factors. The coagulation cascade involves a series of enzymatic reactions, where each of the participating molecules in the coagulation process is referred to as a "factor." These coagulation factors are commonly identified by using a designated Roman numeral chosen according to the order in which they were discovered and with an “a” lowercase to indicate the active form [140]. In these reactions, a zymogen (an inactive enzyme precursor) and its glycoprotein cofactor are activated to become active components that then catalyze the subsequent reaction in the cascade. An active enzyme cleaves a portion of the next inactive protein in the cascade (activating it), ending in the formation of cross-linked fibrin [141], as shown in Figure 4.

When new compounds with anticoagulant potential are evaluated, their ability to prevent the action of coagulation factors is determined, thereby blocking the cascade almost from its inception [135,142]. Each component must function correctly and sufficiently in the process to ensure clot formation. If one or more coagulation factors are deficient, or if one or more are not working correctly, the clot may not form and bleeding may not be controlled [140].

Anticoagulants make it difficult for the blood to coagulate, thereby preventing the formation of clots or their growth and favoring their disappearance [141]. When an injury occurs to a blood vessel or tissue within the body and bleeding occurs, the body initiates a process of clot formation at the site of the damage to help stop the bleeding; this mechanism is known as hemostasis [141]. During this process, platelets stick together and become activated at the injury site. In parallel, the coagulation cascade is initiated, and coagulation factors including fibrinogen are also activated [143]. Fibrinogen is converted by thrombin into insoluble fibrin strands that cross each other, forming a fibrin network that adheres to the main focus of the lesion. Thus, a stable clot prevents further blood loss with platelets and remains in place until the wound heals [140].

Heparin is a sulfated polysaccharide commonly used as an anticoagulant. It acts by inactivating thrombin and activating factor X (factor Xa) through an antithrombin (AT)-dependent mechanism [144]. Despite being one of the most common anticoagulants, heparin has some drawbacks, such as the risk of contamination with oversulfated chondroitin sulfate, which can cause death due to its severe hypotensive effects [145]. In addition, adverse reactions to heparin include hemorrhage, heparin-induced thrombocytopenia, osteoporosis, general hypersensitivity reactions, and elevations of aminotransferase levels [146]. Notably, the number of molecules with anticoagulant activity isolated from natural sources has increased in recent years [139]. Some prominent examples include the anticoagulant active fraction (AAFCC) purified from the leaves of *C. colebrookianum* [147], the ethanolic extract of *Mikania laevigata* [142], the water-soluble polysaccharide fraction (GSP-3) from *Gentiana scabra* bunge roots [148], the active components obtained from Safflower [135], and the sulphated polysaccharide isolated from *Globularia alypum* L. [149].

Coagulation tests such as prothrombin time (PT) and activated partial thromboplastin (aPTT) methods are used in vitro to assess the influence of compounds on blood plasma coagulation [150]. PT, aPTT, and thrombin time (TT) tests can inhibit blood coagulation through the intrinsic, extrinsic, and common pathways of the thrombin cascade and allow an assessment of coagulation, respectively [151]. The intrinsic and extrinsic pathways are two separate pathways that lead to the formation of a blood clot. The main difference between the intrinsic and extrinsic pathways in blood coagulation is that the former pathway is activated after exposure to endothelial collagen, which is only exposed when endothelial damage occurs. In contrast, the second pathway is activated through tissue factor release by endothelial cells after external damage [140,141]. As shown in Figure 5, these three tests are mainly used to discover drugs with anticoagulant properties. First, the test compound is incubated with human-platelet-rich plasma using a specific reagent for each type of test. Then, fibrin formation is evaluated through clots, and the test result is determined through an analyzer and expressed in coagulation time (seconds) [142]. To perform anticoagulant activity tests, platelet-rich plasma (PRP) must be obtained from healthy people because contamination by anticoagulant agents and liver diseases can result in false-positive results with the compounds of interest. For anticoagulant evaluation, the parallel use of intrinsic and extrinsic pathway tests (aPTT, PT) is recommended to determine the anticoagulant properties of the compound of interest [150]. The aPTT and PT assays are the most used in vitro tests nowadays due to their low costs and rapidity.

## 7. Antihypertensive Activity

Hypertension or high blood pressure is a complex, multifactorial disease that contributes to morbidity and mortality in industrialized countries [152]. Although numerous preventive and therapeutic pharmacological interventions exist, as of 2021, approximately 700 million people worldwide still suffer from poorly controlled hypertension [153]. Therefore, there is growing interest in finding novel natural or synthetic molecules that are helpful in preventing and treating hypertension.

Angiotensin-converting enzyme (ACE) is one of the primary regulators of blood pressure. It acts via two central mechanisms: (i) converting the decapeptide angiotensin I (angI) into the potent vasoconstrictor (hypertensive) octapeptide angiotensin II (angII) and (ii) catalyzing the degradation of the antihypertensive peptide bradykinin. Angiotensin II increases blood pressure by stimulating a GPCR-activated pathway [154], resulting in the release of Ca^2+^, activation of protein kinase C (PKC), and subsequent vasoconstriction via the inactivation of the myosin-light-chain kinase (MLCK) (Figure 6).

ACE is a dicarboxypeptidyl peptidase, which cleaves off the two terminal amino acids of angI to form angII. Unlike other peptidases, ACE is not specific to a single substrate. Instead, it cleaves several natural peptides such as bradykinin, substance P, and tetrapeptide *N*-acetyl-Ser-Asp-Lys-Pro (Ac-SDKP) [155] and several synthetic substrates. Therefore, the ability of molecules to inhibit the production of angII by ACE is an effective strategy for measuring the antihypertensive activity of isolated plant extracts in vitro (Figure 6).

Most of the methods for measuring ACE activity in vitro rely on the incubation of a reaction mix containing purified ACE and synthetic substrates such as furanacryloyl (FA-PGG) [156], 3-hydroxybutyryl-Gly-Gly-Gly (3HB-GGG) [157], hippuryl-histidyl-leucine (HHL) [158], dansyltriglycyne [159,160], or benzoyl-[l1-^14^C] glycyl-l-histidyl-l-leucine. Each technique uses the same principle, only differing in the output (colorimetric, fluorometric, radioactive, etc.). The ACE activity or inhibition is determined by adding the substrate and the potential inhibitor compound and comparing the activity to a sample without the inhibitor molecule or extract. One of the advantages of using ACE activity tests to evaluate the activity of potential antihypertensive agents is that due to the central role of ACE in controlling blood pressure in vivo, finding novel ACE inhibitors can directly impact the development of antihypertensive agents. The assay has been used since the 1960s and different substrates and outputs have been described; it can be adapted to evaluate the activity of a vast, diverse array of isolated molecules or mixtures.

Many of the original ACE tests require several intermediate reactions and laborious sample preparation steps (purification, extraction, etc.) to finally read the test result, making the assay hard to automate or increase its throughput. Improvements have been made in the last decades, and the current methods are more straightforward, cheaper, and faster, allowing the development of some high-throughput ACE assays [161,162]. Despite these efforts, given the enormous impact of hypertension as a global health concern, there is still room for further improvements in assay parameters, such as by developing simpler readouts and high-performance assays or reducing the overall cost and enhancing the robustness of ACE activity tests.

## 8. Antioxidant Activity

The antioxidant activity can be described as the property of a given compound that in low concentrations can inhibit or decrease the oxidation of its substrate [163]. An antioxidant acts on free radicals that negatively affect biological systems by neutralizing them through electron donation or by breaking down processes [164]. This inhibition or neutralization can be measured in vitro through chemical (e.g., ABTS, DPPH, FRAP, ORAC, CUPRAC) and biochemical methods (e.g., oxidation of low-density lipoprotein (LDL) assay, thiobarbituric acid reactive substances (TBARS) assay). Nevertheless, the first ones are preferred over the second ones due to their simplicity, speed, and low costs. Therefore, this section is focused on the most frequently used chemical-based methods to evaluate the antioxidant activity of natural molecules in vitro (Table 5). For a complete review of the antioxidant methods, please refer to [164,165,166,167]. Several metabolites and phytoconstituents have shown antioxidant properties because of their scavenging ability; these include phenolic compounds, flavonoids [168,169], anthocyanins [170], and polysaccharides [171]. Many of these molecules are commonly present in natural sources, making the study of the antioxidant properties a must in almost every study about the biological activities. Chemical-based assays fall into two categories: hydrogen atom transfer (HAT) reaction-based and single-electron transfer (ET) reaction-based assays. However, some combine the two mechanisms (i.e., ABTS, DPPH). Notably, several methods (i.e., ABTS, DPPH, FRAP) have limited biological relevance because they measure the inhibition of radicals that do not exist in biological systems (e.g., DPPH^•^, ABTS^•+^) or the reducing capacity based upon one specific ion (Fe^3+^) [165]. The exception to this is the oxygen radical absorbance capacity assay (ORAC), which monitors the inhibition of the biologically relevant peroxyl radical (ROO^•^), and chemiluminescence, which can detect the quenching capacity of different oxygen and reactive nitrogen species [172]. The advantages and disadvantages of the chemical methods are described in Table 5 and should be considered when assessing the antioxidant activity and selecting the best method for a given sample. Overall, the selection of a specific antioxidant method is mainly based on its simplicity, cost, and type of sample. Nevertheless, currently the coupling of many chemical assays to high-throughput or automated systems (i.e., HPLC) facilitates the analysis and saves time [173,174], which makes the parallel use of more than one method possible. It is recommended that the antioxidant activity of natural products be evaluated using assays of biological relevance or complementary assays (e.g., chemical and biochemical) to eliminate possible over- or underestimations due to the absence of a biological system and its components [175]. However, the use of methods of biological irrelevance could provide insights into the antioxidant properties and could guide the research towards more appropriate quantification methods, including in vivo analyses. The future research is focused on developing electrochemical sensors and biosensors, which could be employed for the rapid quantification of complex samples in small amounts and could provide additional information about the kinetics and mechanisms involved in the antioxidant activity [176].

## 9. Antimicrobial Activity

The usefulness of plant extracts for antimicrobial therapy has been promising since ancient times [191]. However, over the last few years, about 250,000 higher plant species have been described, 17% of which have been evaluated according to biological aspects and only 10% which been assessed for antimicrobial activity and several classes of secondary metabolites (alkaloids, flavonoids, saponins, etc.) [192].

Antimicrobial resistance (AMR) is a significant threat to human health worldwide. In 2019, an estimated 4.95 million deaths were associated with AMR, of which 1.95 million were deaths associated with infections caused by fungi, mainly *Candida, Aspergillus,* and *Cryptococcus* [193]. Among the bacteria, the leading pathogens for fatalities related to antibiotic resistance are *Escherichia coli*, *Staphylococcus aureus*, *Klebsiella pneumoniae*, *Streptococcus pneumoniae*, *Acinetobacter baumannii*, and *Pseudomonas aeruginosa.* The synergistic effect of antibiotics with plant extracts against resistant bacteria may lead to new options for treating infectious diseases when the antibiotic is no longer effective during treatment [194].

The Clinical Guidelines of the European Committee on Antimicrobial Susceptibility Testing (CLSI) and the European Committee on Antimicrobial Susceptibility Testing (EUCAST) [195,196] seek to standardize testing for clinical purposes; however, when it comes to natural extracts, these guidelines are also used since there are no standards of their own [197,198,199]. Therefore, it is essential to consider the limitations that this entails. For example, natural extracts may not be hydrophilic, and there is no database of minimum inhibitory concentrations for mixtures of molecules to compare results. This lack of standardization regarding the techniques used to determine antimicrobial activity does not always allow the results to be compared between investigations focused on natural products [200].

Although new technologies have emerged to assess antimicrobial activity, traditional technologies are the most widely used for bacteria and fungi. For example, in vitro antimicrobial evaluation methods emerged in the 1960s and responded to rapid, simple tests that evaluate various plant extracts or pure compounds at a low cost [201]. These methods involve the diffusion of the potential antimicrobial compound through solid or semi-solid culture media to inhibit the growth of sensitive microorganisms (Figure 7) [197]. The reference methods are disk diffusion and broth or agar dilution assays; however, in the last decades, new evaluation methods have emerged to overcome some of the disadvantages of traditional methods, such as the response time, low sensitivity, and reproducibility [202,203]. An overview of the commonly used susceptibility testing methods is shown in Table 6.

The research into natural compounds has shown significant progress in discovering new molecules with antimicrobial activity. The primary natural compounds with valuable antimicrobial activity are medicinal plants and microorganisms. However, given the great diversity of compounds with antimicrobial potential, it is necessary to have adequate tools to facilitate screening, reduce costs, and obtain rapid, quality results [232].

According to the CLSI and EUCAST guidelines, disk diffusion assays are considered the gold standard for evaluating antimicrobial activity. However, in the case of natural extracts, there is no adequate method in all cases, so the decision must consider, among other aspects, the extract’s polarity and complexity, the available equipment, and the possibility of automation [233]. The emergent methodologies to assess the antimicrobial activity of natural extracts include isothermal microcalorimetry (IMC) and mass spectrometry. However, in both cases, these methodologies are expensive and require specialized equipment and training, which limits their use broadly [234,235,236].

In addition, the emergence of new antimicrobial resistance mechanisms requires that the performance of susceptible devices be constantly reassessed and updated periodically, along with new automated instruments that can provide faster results, save money, and reduce the labor requirements [205,237]. Hence, further improvements in the currently used and novel antimicrobial susceptibility test (AST) methods and instruments are mandatory to speed up the determination of antimicrobial efficacy in clinical and research microbiology laboratories in the foreseeable future.

## 10. Conclusions

The high structural and physicochemical diversity of natural molecules produced by plants, microorganisms, and insects makes them a virtually inexhaustible source with potential for the discovery and development of novel bioactive compounds with applications in the pharmaceutical, food, and biotechnological industries, to name only a few [238,239]. Additionally, the selectivity, specificity, and potency are common features of many natural compounds as the result of their ongoing evolutionary specialization over millions of years. Given the immense variety of biological molecules and the high array of promising activities and evaluation methods, it is crucial to establish the most relevant tests for the initial screening stage [234]. 

The first step in the biological evaluation of natural molecules commonly involves evaluating the compound in a range of in vitro biochemical and pharmacological assays. These experiments aim to establish a solid grounding regarding the compound’s properties and to understand its mechanism of action. This information is then used to select the most suitable compounds and evaluate their activity in more complex in vitro and in vivo assays [240]. The selection of the methods to be used must be made on a case-by-case basis, considering their advantages and limitations. Additionally, the use of complementary assays during the evaluation of biological activities to avoid underestimating their potential activity and to aid with identifying the pharmacological mechanisms is highly recommended. The future trends in assessing the biological activity of natural molecules are focused on establishing simplified and automated protocols that enable high-throughput screening [235], facilitating subsequent analyses, shortening the evaluation time, and allowing the parallel use of multiple methods [236]. Moreover, different systems are being developed to support more physiologically relevant data collection and enable real-time monitoring.

It is also important, after the initial identification of the biological potential from an extract or a natural source, to determine the specific molecule responsible for the given activity. Thus, this initial screening can be further complemented by identifying specific metabolites or active molecules present in the sample through approaches such as fractionation, NMR, HPLC-MS/MS, high-resolution Fourier transform mass spectrometry (FTMS), and photo-diode arrays. Notably, the data generated from MS and NMR are large and complex. Therefore, several computational approaches and multivariate analysis techniques such as the cluster analysis and partial least squares regression (PLS) methods, as well as the principal component analysis (PCA) method, are employed to identify and discriminate the metabolites present in each sample [241]. Another approach is to depict the interactions between metabolites and biological pathways as networks. Here, the metabolite is placed in a node and linked to a metabolic interaction. KEGG, Human Recon 1, and EHMN are some of the existing metabolic databases focused on metabolic pathways [242]. Moreover, high-throughput data can be used to identify pathways through Bayesian-based networks, correlation-based measures, and Gaussian graphical models (GGMs) [243,244]. Overall, complementary strategies should be employed to extract as much information as possible from an experiment and to help develop novel therapeutics.

Finally, as discussed throughout this review, it is worth mentioning that identifying natural products with biological activities is crucial for drug discovery. However, it is also important to recall that regardless of whether the compounds are obtained from natural sources or chemically synthesized, they are not exempt from potentially causing adverse reactions. Therefore, their adverse effects must be evaluated with the same rigor required when evaluating their biological activity.

## Figures and Tables

**Figure 1 molecules-27-04490-f001:**
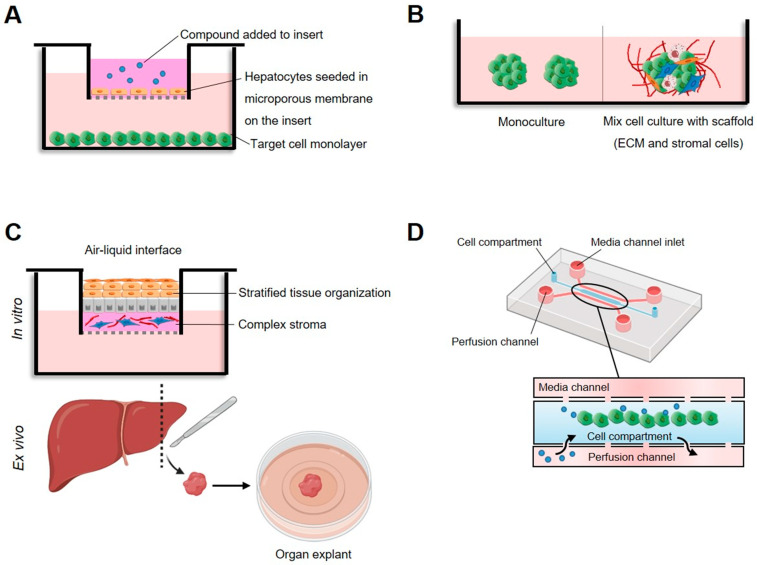
High-tech in vitro models to assess cytotoxicity in cultured mammalian cells. (**A**) Dual chamber, test compound, and metabolites diffuse through the microporous barrier toward target cells. (**B**) Three-dimensional cellular models based on multicellular spheroids or organoids consisting of target cells or the co-cultivation of several types of cells on extracellular matrix (ECM). (**C**) Organotypic cultures, whereby cells, organ slices, or whole organs are cultured on a tissue culture insert that is either submerged in medium or maintained at an air–liquid interface to ensure sufficient oxygen supply. (**D**) Microfluidic system based on a mixture of cells and matrix collected in the central channel and medium flowing from the lateral channels that keeps particles in homogenous suspension.

**Figure 2 molecules-27-04490-f002:**
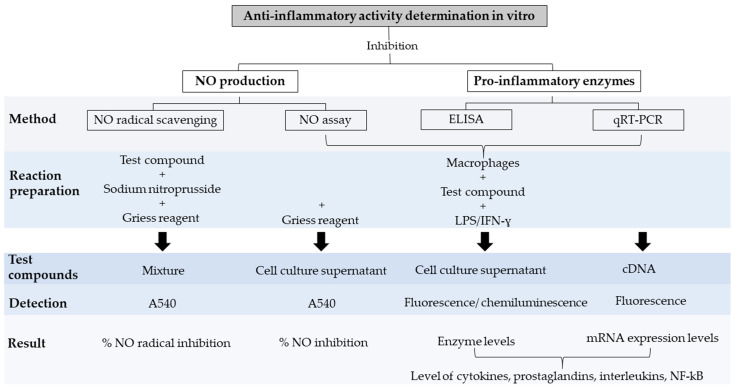
Strategies employed to assess anti-inflammatory activity in vitro. Nitric oxide (NO), lipopolysaccharide (LPS), interferon-gamma (IFN-γ), nuclear factor kappa B (NF-κB).

**Figure 3 molecules-27-04490-f003:**
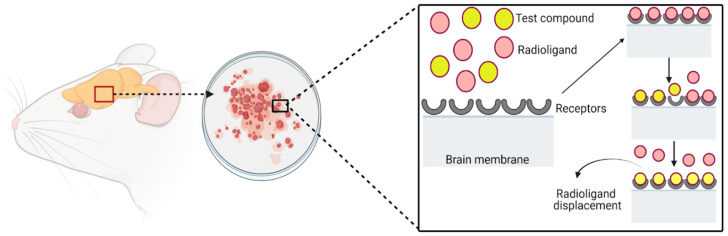
Sites in the rat brain tissue, where both opiate antagonists and agonists compete for the same receptors, opiate potencies, and antagonists in displacing 3*H*-naloxone binding parallel to their pharmacological potencies.

**Figure 4 molecules-27-04490-f004:**
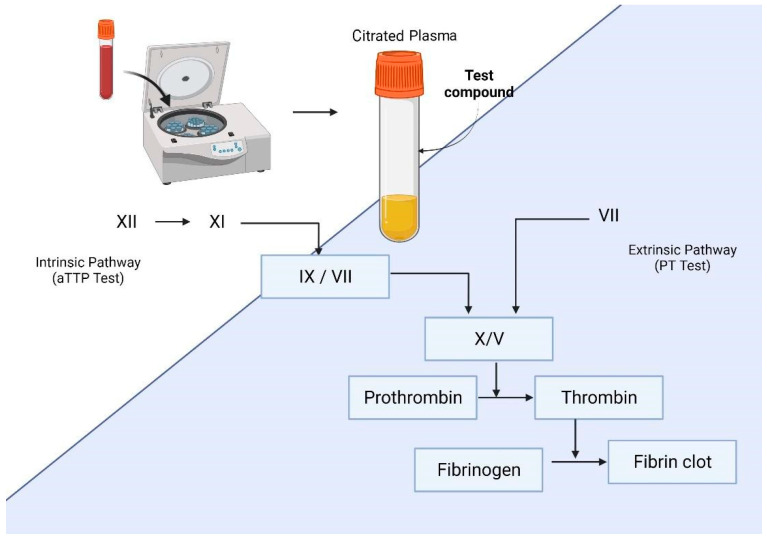
The activation of coagulation factors in vitro for clot formation by adding the test compound as a possible therapeutic agent.

**Figure 5 molecules-27-04490-f005:**
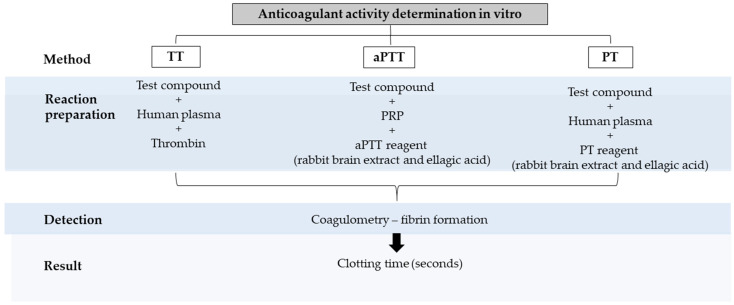
Strategies employed to assess analgesic activity in vitro. TT: thrombin time; aPTT: activated partial thromboplastin; PT: prothrombin time; PRP: platelet-rich plasma.

**Figure 6 molecules-27-04490-f006:**
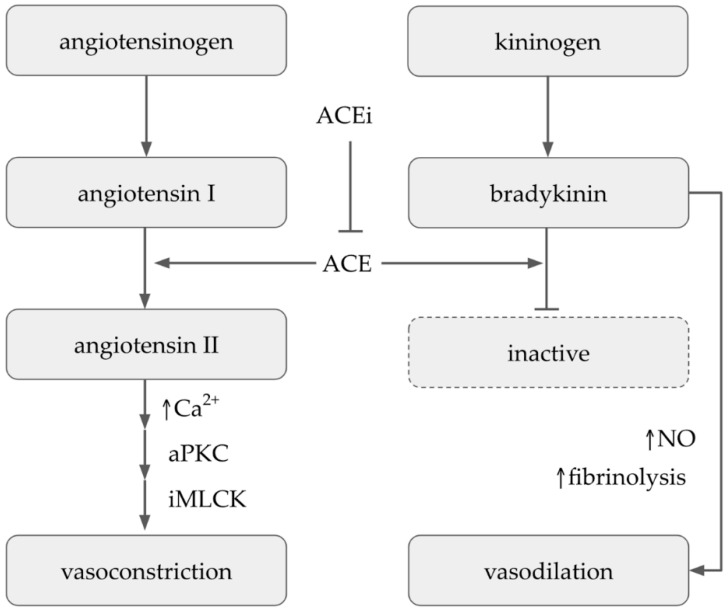
The inhibition of angiotensin-converting enzyme. Calcium (Ca^2+^), angiotensin-converting enzyme (ACE), angiotensin-converting enzyme inhibitors (ACEi), atypical protein kinase C (aPKC), myosin-light-chain kinase (MLCK), nitric oxide (NO).

**Figure 7 molecules-27-04490-f007:**
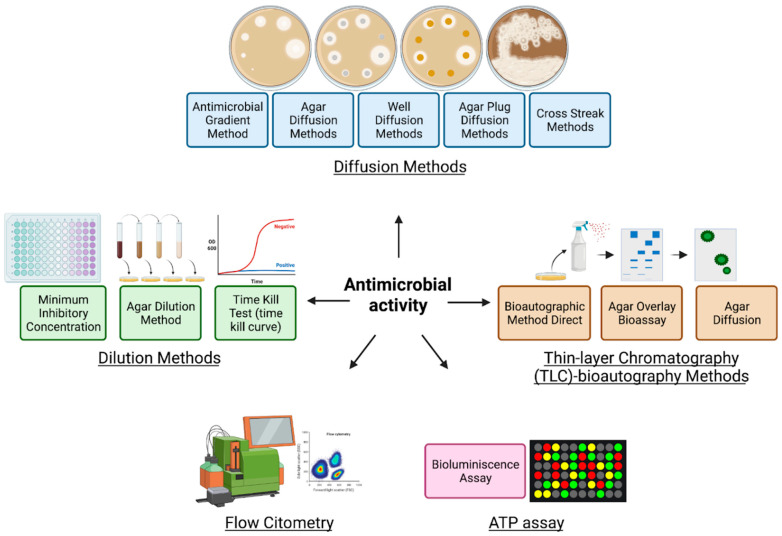
Summary of the most relevant antimicrobial activity methods.

**Table 1 molecules-27-04490-t001:** Most common methods for the determination of cytotoxicity in cultured mammalian cells.

Method	Principle	Detection	Advantages	Disadvantages	Ref.
**Cell Viability and Proliferation**					
Dye exclusion	Detection of plasma membrane integrity	Trypan blue colorimetric/H + M	- Simple - Low-cost - Immediate readout - Widely available - Not affected by enzyme changes/activity	- Not suited for large numbers of samples - Limited sensitivity: dead cells vs. live damaged cells - It may not detect cell injury - Dye uptake estimate can be subjective - Toxic to mammalian cells - Possible counting errors (~10%): poor dispersion/dilution of cells, cell loss during cell dispersion, air bubbles, etc.	[27,28,29,30]
		Crystal violet colorimetric/M, PR	- Simple - Rapid - Reliable - Sensitive - Economical - Can be used under different conditions - Additive or synergistic interactions (metabolism-independent)	- Does not detect changes in cell metabolic activity - Not suitable for analyses featuring affected cell metabolism compounds - Not suited to determine cell growth rate	[13,31]
Metabolic activity	Detection of mitochondrial dehydrogenase and oxidoreductase activity	Tetrazolium salts (MTT, XTT, MTS, WST) colorimetric/PR	- Easy to use - Sensitive - Safe - High reproducibility - Economical - PUsed for large samples	- Hours to readout - High background (interference with reagent/media) - Additional control experiments needed to reduce false-positives/-negatives - Reduction is affected by metabolic and other factors - Incubation time, concentration, metabolic activity, can affect the final reading	[27,31,32,33,34]
		Resazurin fluorescent/PR	- Inexpensive - More sensitive than tetrazolium assays - Can be multiplexed with other techniques	- Possible high background (interference with reagent/media) - Hours to readout - Fluorescent interference - Close cell–cell interactions affect uptake - Direct toxic effects on the cells	[35,36,37]
Energy metabolism	Correlation between a bioluminescent reaction and the ability to synthesize ATP	Luciferase and luciferin luminescent/PR	- Immediate readout - Sensitive - Deficient background - Stable luminescent signal - Useful to detect cellular death in a mixed cell culture model - Does not need an incubation step	- Limited repeatability - Endpoint - Needs cell engineering - Decrease in luminescent signal with increased cell death - Difficult to distinguish small changes in the number of dead cells	[14,27]
Enzyme release-based	Determination of plasma membrane integrity (LDH release)	Lactate + tetrazolium salts/fluorescent probe colorimetric, fluorescent/PR	- Reliable - Quick - Simple - Non-destructive measurement - The culture medium can be used for analysis	- High background - Limited to serum-free or low-serum conditions - Low EC - Difficult to detect low cytotoxic effects - High variability	[27,32,38]
Colony formation	Determination of clonogenic growth	Low-adherence plates/M	- Low interference - Colonies can be counted without being stained	- Time-consuming and labor-intensive - High intra-individual variability - Limited to adherent cells - Restricted by low cell density conditions and growth factors - Stress may affect cellular repair - Colonies may be lost during washing and staining - Overestimate cell damage/death - Overly sensitive threshold - Low sensitivity range	[38,39,40,41]
**Cell Cycle and Apoptosis**					
Cell cycle arrest	Distribution of cell population in each cell cycle phase	PI, 7-AAD fluorescent/FC	- Simple - Single-cell quantification of stained DNA - Bleft fluorescent signal	- Endpoint (fixed-permeabilized cells) - Does not detect floating cells	[42,43]
Apoptosis/ necrosis	Detection of membrane integrity	PI/7-AAD, annexin V fluorescent/F	- Simple protocols - Short incubation time - Economical - Stable	- Difficult to differentiate between living and dead fixed cells - Cells are continuously dying in the sample - Confirmation by other methods needed to avoid false-negative bias	[44,45,46]

H: Hemocytometer, M: microscopy; PR: plate reader; FC: flow cytometry; LI: light imager; T: thermocycler; GE: gel electrophoresis; PI: propidium iodide; 7-AAD: 7–aminoactinomycin D; MTT: 3-(4,5-dimethylthiazol-2-yl)-2,5-diphenyltetrazolium bromide); XTT: (2,3-bis(2-methoxy-4-nitro-5-sulfophenyl)-5-carboxanilide-2*H*-tetrazolium); MTS: 3-(4,5-dimethylthiazol-2-yl)-5(3-carboxymethonyphenol)-2-(4-sulfophenyl)-2*H*-tetrazolium; WST: water-soluble tetrazolium salts; EC: electric conductivity.

**Table 4 molecules-27-04490-t004:** Enzyme inhibition method for the determination of the analgesic activity of molecules.

Method Type	Detection (Output)	Description	Advantages	Disadvantages	Ref.
Enzyme inhibition	Fluorometric detection using fluorogenic peptide DANGPG	- Detects the inhibition of the degradation of enkephalinase. This enzyme uses DANGPG as substrate and cleaves the peptide bond of DANGPG leading to a fluorescence increase - Enkephalinase induces inactivation of ANF. The protection of endogenous ANF against inactivation may result in analgesic applications - Hazards of handling high levels of radioactivity - Requires a certain level of expertise	- Highly sensitive test - Quantitative data - Rapid	- High cost - Susceptible to different fluorescence interferences	[131,135,136]

DANGPG: dansyl-*D*-Ala-Gly-Phe(*p*NO_2_)-Gly; ANF: atrial natriuretic factor.

**Table 5 molecules-27-04490-t005:** Common chemical methods for determining the antioxidant activity of natural molecules.

Method Name	Description	Detection Method	Advantages	Disadvantages	Ref.
**Radical/ROS-based scavenging assays**					
2,2′-azinobis (3-ethylbenzo-thiazoline 6- sulphonate) (ABTS)/Trolox equivalent antioxidant capacity (TEAC) test	- HAT/ET - Antioxidant reaction with an organic cation radical - ABTS is converted to its radical cation by addition of sodium or potassium persulfate - The ABTS^•+^ radical loses absorption at 734 nm if reduced by an antioxidant	Spectrophotometry (A_734_)	- Rapid - Cheap - Simple - Can be used over a wide range of pH values - Can be coupled with online HPLC - Used for hydrophilic and lipophilic antioxidants	- Limited relevance to biological systems - Difficulties with the formation and stability of colored radicals - Phenolic compounds with low redox potentials can react with ABTS^•+^	[167,177,178,179]
*N*, *N*-diphenyl-*N*’-picrylhydrazyl (DPPH) free radical	- HAT/ET - Antioxidant reaction with an organic radical - The DPPH^•^ free radical loses absorption at 515-517 nm if reduced by an antioxidant or a free radical species	- Spectrophotometry (A_515_) - EPR - Amperometric detection	- Rapid - Cheap - Simple - Stable at room temperature - Used for hydrophobic antioxidants - Can be coupled with online HPLC	- Limited relevance to biological systems - Difficulties with the formation and stability of colored radicals - Not recommended for samples with anthocyanin leads - Could be interfered by borate presence	[174,177]
Oxygen radical absorbance capacity (ORAC)	- HAT - Monitors the inhibition of peroxyl radical-induced oxidation - Requires peroxyl radical generators - The peroxyl radical reacts with a fluorescent probe resulting in the loss of fluorescence	Fluorometry	- Considered to be of biological relevance - High-throughput assay possible	- Peroxyl radical formation is thermosensitive - Could be interfered by hydroxyl radical scavengers and metallic ions - Β-phycoerythrin (fluorescent probe), may show inconsistency from lot to lot and photo instability	[175,180,181,182]
Chemiluminescence	- HAT - Consists of a chemiluminescent species, an oxidant (hydrogen peroxide) in the presence or absence of a metal or enzymatic catalyst, and an antioxidant or extract - Decreases of chemiluminescence intensity as a result of the antioxidant	Fluorometry	- Rapid - Cheap - Sensitive - Robust - Stable - It has been automated - Can be coupled with online HPLC	- Limitations with luminol-based antioxidant assays - Requires pH > 8.5	[178,179,180,183,184]
**Non-radical redox-potential-based assays**					
Ferric reducing antioxidant power (FRAP)	- ET - Measures the reduction (Fe^3+^)–ligand complex to (Fe^2+^)–complex by antioxidants - Ligand used to facilitate detection: TPTZ - Antioxidant activity is determined as an increase in absorbance at 593 nm	- Spectrophotometry (A_593_) - Electrochemical (coulometric titrants)	- Rapid - Cheap - Simple - Modified to allow the measurement of diverse sample types - Used for hydrophilic antioxidants - FRAP test by coulometric titration is extremely sensitive and reliable - Can be coupled with online HPLC	- Limited relevance to biological systems - Redox chemistry of ferric ion involves slower kinetics than the copper ones - Not sensitive toward thiol-type oxidants - Requires acidic pH (pH 3.6)	[173,178,179,185,186,187]
Cupric reducing antioxidant capacity (CUPRAC)	- ET - Measures the reduction Cu^2+^ to Cu^+^ by antioxidants - Ligand used to facilitate detection: neocuproine - Reduction in Cu^2+^–neocuproine complex to Cu^+^–neocuproine has an absorption peak at 450 nm	Spectrophotometry (A_450_)	- Simple - Stable - Sensitive - Favorable redox potential - Used for hydrophilic and lipophilic antioxidants - Can be coupled with online HPLC - High-throughput use possible	- Takes longer time to measure complex mixtures compared to other methods - Resulting product is more unstable than in other methods - Possible interference of absorption spectra between the oxidizing agent and the studied compound	[164,166,188,189,190]

HAT (hydrogen atom transfer), ET (single-electron transfer), EPR (electron paramagnetic resonance), HPLC (high-performance liquid chromatography), TPTZ (tripyridyltriazine).

**Table 6 molecules-27-04490-t006:** Traditional and emerging methodologies used to assess the antimicrobial activity of natural products.

	Method Name	Description	Advantages	Disadvantages	Ref.
Diffusion methods	Agar diffusion method	The antimicrobial agent diffuses from disks or strips into the solid culture medium that has been seeded with a pure culture	- Low cost - Rapid and time-saving - Ability to test enormous numbers of microorganisms and antimicrobial agents - Ease to interpret obtained results	- Does not work on fastidious bacteria - Qualitative - Does not distinguish bactericidal/fungicidal and bacteriostatic/fungistatic activity - Disc/well preparation is time-consuming - No automation available	[198,199,204]
	Well diffusion method	Diffusion of a liquid antimicrobial agent placed in a well punched into the solid culture medium that has been seeded with a pure culture			
	Agar plug diffusion method	The first bacterium or fungus is grown on agar plates, here it will secrete molecules that diffuse in the agar which then is cut and placed on another agar plate inoculated with a different microorganism	- Low cost - Simple - Highlights the antagonism between microorganisms	- Time-consuming - No automation available	[199,200,201]
	Antimicrobial gradient method (Etest)	Based on creating a concentration gradient of the antimicrobial agent tested in the agar medium where it is exposed to the selected microorganism	- Quantitative - Used for MIC determination - Simple	- Expensive when compared to other diffusion methods	[205,206]
	Cross streak method	The first microbial strain is streaked in the center of the agar plate and incubated then in the same plate is seeded the second microorganism by a single streak perpendicular to the central streak	- Simple - Rapid screening - Identifies antagonism between microorganisms	- No quantitative - Margins of the zone of inhibition are usually very fuzzy	[207,208]
Thin-layer chromatography (TLC)–bioautography methods	Bioautographic method direct	The antimicrobial activity is assessed directly onto the TLC plates where the extracts are separated by chromatography across a TLC plate, then the microorganisms are also applied by spray-identifying the localization of the fraction with antimicrobial potential	- Works for fungi and bacteria - Consistent with spore-producing fungi - Fast and cheap - Simple	- Difficulties in obtaining complete contact between the agar and the plate - Consistent with spore-producing fungi - Fast and cheap - Simple	[209,210]
	Agar overlay bioassay	The TLC plate is covered with agar seeded with the test microbe and the antimicrobial compounds are diffused onto the agar medium	- Provides well-defined growth inhibition zones - Not sensitive to contamination - Fast and cheap - Simple	- Time-consuming - Low sensitivity	[211,212,213]
	Agar diffusion	The antimicrobial agent is transferred from a TLC to an agar plate previously inoculated with the test microorganism	- Fast and cheap - Sensitive - Works with bacteria and fungi	- Agar is prone to adhere to silica gel due to the prominent adsorption between them - Compounds will be lost during the transfer from the thin-layer plate to the culture medium	[212,214]
Dilution methods	Broth dilution method Minimum inhibitory concentration (MIC) determination	Uses tubes or microdilution plates to measures the lowest concentration of antimicrobial agent that completely inhibits the growth of the bacteria/fungi	- Quantitative - Good reproducibility	- Manual task of preparing the antibiotic solutions for each test	[215,216]
	Agar dilution method	Similarly, to the procedure used in the disk diffusion method, a desired concentration of the antimicrobial agent is placed into an agar medium	- Suitable for both antibacterial and antifungal susceptibility testing	- If not automated, very laborious	[217,218,219]
	Time-kill test (time-kill curve)	Based on a time/concentration-dependent analysis of antimicrobial effects. Several tubes containing varying concentrations of the antimicrobial agent are seeded with the bacteria/fungi and the percentage of dead cells is determined along with the assay	- Can be used to determine synergism or antagonism between drugs - Suitable to identify bacteriostatic/fungistatic and bactericide/fungicide effects	- Time-consuming	[220,221]
ATP bioluminescence assay	Bioluminescence ATP based	Based on the capacity to measure adenosine triphosphate (ATP) produced by bacteria or fungi	- Rapid and easy - Quantitative - In situ evaluation	- Expensive - Difficult to differentiate the microbial ATP from other organic debris - Adapted only for solid surfaces	[222,223,224]
Flow cytofluorometric method	Flow cytofluorometric	Based on the capacity of damaged cells to emit a positive signal that is detected by flow cytometry analysis. The microorganism is exposed to antimicrobial agents and then stained with the intercalating agent propidium iodide.	- Three subpopulations (dead, viable, and injured cells) can be clearly discriminated - High-throughput screening	- More expensive - Flow cytometry equipment is required	[225,226,227]
Biofilm inhibition	Microtiter plate biofilm production assay	Based on the potential of the extracts to prevent initial cell attachment to microtiter plates. Biofilm formation/inhibition is observed through (OD) of the crystal violet present in the destaining solution measured at 595 nm	- Rapid - Non-expensive - High-throughput screening	- Qualitative - Low sensitivity	[228,229]
Quorum sensing inhibition	Violacein quantification	The inhibitory activity is measured by quantifying violacein production in a microplate reader using a spectrophotometer at 585 nm where the microorganism and extracts are cultured at different concentrations.	- Simple - Rapid - Inexpensive	- The color of the extracts can interfere with the OD measurement.	[230,231]

ATP: adenosine triphosphate; OD: optical density.

## Data Availability

All tables were created by the authors. All sources of information were adequately referenced. There was no need to obtain copyright permission.

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
