# Peer review of "Evaluation of Biological Activity of Natural Compounds: Current Trends and Methods"

_molecules, 2022, doi:10.3390/molecules27144490_

Round 1

Reviewer 1 Report

The authors aim at providing an exhaustive list of in vitro and in vivo assays in order to test natural bioactive molecules and suggest minimum techniques essential for correctly assessing molecules with biological potential.

If the intention of the authors is laudable and the proposal of pointing (dis)advantages for each method of interest, I was expecting the description of new methods. Most of the described methods are well-known and can be found easily in literature. Moreover, the authors do not talk about the material (plant extract) which is an essential point. What is the advantage of testing a whole extract compared to a bioguided fraction. How deciphering the plant composition in the perspective of isolation of an active compound and a future lead compound? What is the place of plant metabolomics and molecular network to either focus on unknown molecules or to discriminate molecules in high amount that could explain the pharmacological activity.

Another major point is that authors justify the research of new natural bioactive compound because of “the lack of therapeutic options to treat various diseases” (abstract). However, authors described method assessing anti-hypertensive, anti-hyperglycemic, anti-inflammatory, analgesic, and anticoagulant activities; pharmacological area for which we have many therapeutical solutions… Testing bioactive molecule in this context is not the best illustration of what they said.

Finally, at the end of the review, authors propose a minimal list of assay to assess bioactive molecules in the respective pharmacological area of interest. I am disagreed with this list as for major pharmacological area, the proposed minimal test cover only a little part of pharmacological mechanisms responsible for the activity. For example, for anti-hypertensive drug, they propose to test the activity only on part of the renin-angiotensin system excluding orthosympatic system, diuretics effect, calcium transport… 

Reviewer 2 Report

The submission is a fine and useful review about procedures to study biological effects of natural compounds. Many readers of Molecules will find valuable information and bibliography in their search for new potential drugs. 

In the present context, still heavily affected with COVID-19 pandemy and other diseases caused by viruses (dengue, monkeypox, for example), it would be desirable that the paper ended not with a section "Antibacterial", but with a section "Antimicrobial", and inclusion of procedures for antifungal and antiviral effects. But the present comment is not meant to disqualify the submission.

Minor modifications are needed before final acceptance of the paper:

1. Lines 77 and 112 need rephrasal.

2. Line 119: correct "taste compound" to "test compound"

3. The phrase starting with line 207 and ending with line 210 is too long; it contains two sentences, the second one, starting with "are an excellent source..." has no subject. Correction is recommended to improve grammar and clarity.

4. Line 220: possibly, the text could be improved to "Thus, a compound inducing low NO levels has anti-inflammatory potential".

5. The phrase contained between lines 452-456 is too long. It is recommended to divide it in two phrases, the second one starting with "Nevertheless, the first...".

6. Line 451. Antioxidants involve not only free radical scavengers, but also electron donors, which may be evaluated by the FRAP method. 

Reviewer 3 Report

The Authors of manuscript presented extensive review about different methods (assays) and trends which are use to evaluation biological activities such as cytotoxicity, antihyperglycemic, anti-inflammatory, analgesic, anticoagulant, antihypertensive, antioxidant and antimicrobial activity. It is really interesting and voluminous work.

I have only a few minor comments:

Lines 246, 488, 523: no dot at the end of the sentence;

Lines 396 and 410: ‘Ca2+, ‘2+’ should be in top index;

Sentences in lines 391-397 and 405-411 are the same. Please improve it.

Line 435: “ii) Versatility.” What the Authors had in mind ?

Line 471: Fe3+, 3+ should be in top index;

Table 5, ABTS assay, column 4 – two dashes are without information, please improve it;

Line 506-508: the names of microorganisms e.g. Escherichia coli, etc. should be italic;

Line 544 and 571: Figure 7 is in both lines, in line 571 should be Figure 8.

Line 664: The citation is underline

My suggestion to present the tables on the page in landscape orientation for better readability.

Reviewer 4 Report

Comments and Suggestions for Authors

Manuscript ID: molecules-1748298

Title: Evaluation of biological activity of natural compounds: current 3 trends and methods

Comments:

In the manuscript entitled “Evaluation of biological activity of natural compounds: current 3 trends and methods, Carlos Barba-Ostria and colleagues reviewed the main techniques and protocols used to test the biological activities of natural compounds. They reported the techniques for these tests: Cytotoxicity activity in cultured mammalian cells; Antihyperglycemic activity; Anti-inflammatory activity; Analgesic activity; Anticoagulant activity; Antihypertensive activity; Antioxidant activity and Antibacterial activity. However, there are some modifications required to be done before it is accepted for publication. The following are the specific comments on the manuscript:

Specific comments:

The major defect of this study is the debate or argument is not clearly stated in the introduction section. In fact, authors reported the techniques used without critical point of view from the author or from the bibliography: authors have to advice readers and specialist about the best technique to be used in each test described in the review paper.

In the antibacterial activity paragraph: It will be better if it will be changed to Antimicrobial activity to include yeast, molds, viruses also. In this paragraph it will be better to add a subsection related to antibiofilm and anti-quorum sensing evaluation.

 Hence, the contribution of this review can be better. I would suggest the author to enhance the theoretical discussion and arrives your debate or argument.

Round 2

Reviewer 1 Report

I have still comments to the answers provided by the authors.

My major comments are that authors are too enthusiastic regarding the potential of natural products compared to actual pharmacological treatment  The authors should keep in mind that if a natural compound act on a pharmacological system, it will have adverse effect; a drug that have no adverse effects is probably not efficient... In this context, the research of new natural compounds will provide new therapeutical strategies or new molecules in therapeutical pathway already known... But the safety of natural compounds is not an argument as an active compound in vivo will have adverse effects...

There is also a lack of knowledge concerning the pharmacological aspects...

P2, L61: Within this context, the current review presents some of the most common in vitro assays currently used for identification of bioactive compounds in natural sources. --> you do not provide assays for the identification of the compounds but assays for their pharmacological activity in crude extract...

P17, L318: Some of the current analgesic treatments are associated with certain adverse effects such as gastrointestinal irritation, fluid retention, and prolonged bleeding time [124] --> there is a misunderstanding of the difference between anti-inflammatory and analgesic activities... the adverse effect mentioned here are mostly associated to the minor analgesics that act as anti-inflammatory where prostaglandin are responsible for pain. They inhibit COX-1 which could explain the mentioned adverse effects. Analgesic compound treated in this paragraph act on the nociceptive pain which is driven by fiber C and inhibit the modular portal system... These molecules have no effect on COX and do not have this kind of adverse reactions. As a consequence, natural molecules that will act as analgesic like morphine will not have such averse effect as the current molecules on the market... And natural molecules that are active on this system will have other adverse effect and will be not safer!!! Finally, captopril does not act as analgesic but it is ACE inhibitor; please remove. Salicin acts more on COX system so remove too!!!! 

P17, L350: One commonly used anticoagulant is heparin. However, it presents disadvantages such as low bioavailability... and polysaccharides... --> Heparin is a polysaccharide; in this context, pointing the lack of bioavailability of heparin and be enthusiastic because natural product such as polysaccharides will provide new anticoagulant molecules is a little bit exaggerate... Moreover, lot of polyphenol, peptides and flavonoids have a poor bioavailability...

P21, L467: Additionally, the study of specific plant metabolites present in extracts has advanced in the last decades due to the determination of their molecular formulas via approaches such as fractionation, NMR, HPLC-MS/MS, high-resolution Fourier-transform mass spectrometry (FTMS), and photo-diode arrays [170]. Thus, improving the identification of natural sources with antioxidant activity. --> This phrase should not be added here; it has been added in the conclusion and should be address only there. So remove! However it is not appropriate see below...

P32, L568: This initial screening can be further complemented by identifying  specific metabolites or active molecules present in the sample through approaches such as fractionation, NMR, HPLC-MS/MS, high-resolution Fourier-transform mass spectrometry(FTMS) and photo-diode arrays --> The authors propose adding this phrase to answer to the comments of reviewer regarding the metabolomics approach. However the sentence here does not really answer to this remark. Indeed, authors propose classical approach such fractioning and detection technics but metabolomics cover also and mainly statistical data treatments (PCA analysis, molecular network...) to compare extract and highlighting features that are discriminant between extracts and that could be responsible of the pharmacological activity. Please improve this phrase. 
